# Silicone Resin-Based Intumescent Paints

**DOI:** 10.3390/ma13214785

**Published:** 2020-10-27

**Authors:** Maria Zielecka, Anna Rabajczyk, Krzysztof Cygańczuk, Łukasz Pastuszka, Leszek Jurecki

**Affiliations:** Scientific and Research Center for Fire Protection National Research Institute, Nadwiślańska 213, 05-420 Józefów, Poland; arabajczyk@cnbop.pl (A.R.); kcyganczuk@cnbop.pl (K.C.); lpastuszka@cnbop.pl (Ł.P.); ljurecki@cnbop.pl (L.J.)

**Keywords:** intumescent paint, silicone resin, fire protection of steel structures

## Abstract

Silicone resins are widely applied as coating materials due to their unique properties, especially those related to very good heat resistance. The most important effect on the long-term heat resistance of the coating is connected with the type of resin. Moreover, this structure is stabilized by a chemical reaction between the hydroxyl groups from the organoclay and the silicone resin. The novel trends in application of silicone resins in intumescent paints used mostly for protection of steel structures against fire will be presented based on literature review. Some examples of innovative applications for fire protection of other materials will be also presented. The effect of silicone resin structure and the type of filler used in these paints on the properties of the char formed during the thermal decomposition of the intumescent paint will be discussed in detail. The most frequently used additives are expanded graphite and organoclay. It has been demonstrated that silicate platelets are intercalated in the silicone matrix, significantly increasing its mechanical strength and resulting in high protection against fire.

## 1. Introduction

Silicone resins and branched polysiloxanes with very good thermal resistance are widely used as components of coating materials to meet the requirements of different applications. These compounds can be used to coat various materials, including building materials, ceramics, and construction elements. One of the examples of demanding applications are steel and aluminum structures used in construction, which must be adequately protected against fire in order to maintain their load capacity for a specified period of time, enabling the evacuation and protection of the object. Under fire conditions, the temperature of steel structure elements rises very quickly, reaching the limit temperature in which a loss of mechanical properties occurs. The result is deformation of the structural elements and their collapse. Depending on the type of fire source and its intensity and the massiveness of the structural elements, the critical temperature of steel (450–550 °C) can be reached within a few minutes. The reason for the loss of mechanical properties of steel at elevated temperatures may be stresses related to thermal expansion [1]. Steel can also exhibit a steel creep effect when the structure is exposed to the simultaneous effects of elevated temperature and high stresses. Based on detailed studies, it was found that this effect occurs at temperatures above 400–500 °C, depending on the type of steel [2]. Comparing the thermal stability of both hot-rolled and cold-worked reinforcement bars, it was found that up to a temperature of 400 °C, no significant changes in mechanical strength occur [3]. However, at higher temperatures, a clear reduction in the mechanical strength of cold-worked reinforcement bars was observed, being up to 10–15% at 600 °C. In addition, the residual mechanical deformation at 600 °C was 50% for hot-rolled specimens and about 150% for cold-worked bars, which is important for the strength of the structure during a fire. The role of passive protection of steel against fire is to create an insulating barrier that extends the time it takes to reach a critical temperature of steel to allow people to evacuate and firefighters to act. Depending on the type of structure, location of structural elements, and local standards, the required time of fire resistance of the structure is classified in the range from 15 min to 2 h.

Historically, the first way to create an insulating barrier against the inflow of heat was to enclose the steel structure in a concrete cover (thermal conductivity 1–3 kW/(m·K). The next applied solution was the use of spraying cement or gypsum masses filled with light porous material (which can expand under high temperature conditions), e.g., polystyrene granules, perlite, vermiculite, or mineral fiber materials. These solutions had very limited applications due to the weight of the insulating layer and corrosive properties of cement mixtures which require the initial protection of steel.

Good effects in protection of steel structures against fire are achieved through the use of intumescent paints capable of producing a sintered layer during a fire, isolating the steel structure from high temperature. The best protection of the structure can be obtained when the sintered layer is formed in a controlled manner with the formation of a layer with an appropriate thickness from 1 mm to 10 cm [4]. The basic ingredients of intumescent paint are carbonizable material, e.g., di-pentaerythritol, catalyst in the form of mineral acid or ammonium polyphosphate-APP, blowing agent (most often melamine), and a binder [5,6,7] on an organic or mineral basis. The most commonly used organic binders are chlorinated rubbers [8], phosphorus-containing styrene–acrylic copolymers [9], vinyl copolymers [10], epoxy resins (often in hybrid systems with phosphorus compounds) [11], and acrylic latexes [12,13] or epoxy resin-based binders [14]. The paint binder, in addition to its basic function, ensures good adhesion to the substrate throughout the entire process, as well as flexibility of the insulating coating. The binder is also a source of additional carbon in the insulating layer. Plasticizers and fibrous fillers enable the production of a sufficiently thick and mechanically stable foamed layer. All chemical reactions and physical changes during the swelling process must occur at the correct speed in the correct sequence.

Unfortunately, it should be noted that typical organic binders used to make intumescent paints have a number of disadvantages. First of all, organic binders undergo thermal decomposition with the release of toxic gaseous products. It also causes a deterioration of the thermal insulation properties of the coating because the sintered layer is cracked and has insufficient cohesion. An additional factor causing the imperfections of the sintered layer is a softening point that is too low and thermal decomposition of organic binders, which disturbs the formation of this layer [15]. All these factors can damage the coating during a fire and, consequently, give an insufficient protection of the structure [16,17]. Very good protection results for such structures can be obtained thanks to the use of silicone-based intumescent coatings. Silicones with a branched structure, including silicone resins or branched polysiloxanes, are characterized by very good thermal stability, which may facilitate adequate protection, because the loss of mechanical properties of steel usually occurs at a temperature of about 500 °C [18].

At this point, a second inorganic type of intumescent coatings based on alkali silicate [19] should also be mentioned. They are mainly used in the fire protection of wood. These paints swell on exposure to fire mainly due to endothermic loss of hydration water. Additionally, due to their ability to melt, they form a solid rigid foam consisting mainly of hydrated silica. Their use as protective coatings is limited and they are mainly used as firewalls. Therefore, they will not be discussed in this publication.

The aim of this review is to present the most important information concerning silicone fire-retardant intumescent paints based on a literature review made using the following keywords: intumescent paint, silicone resin, fire protection of steel or aluminum structures, silicone intumescent paint application. Literature review was made using the following databases: Web of Knowledge, Scopus, and Google Scholar. The research also covered Espacenet, Patentscope, and Google Patents, which resulted in the presentation of selected patents relevant to the subject. The overview is divided into the following main sections including discussion focused on the effect of silicone resin structure on its thermal stability, the effect of silicone resin structure, and the type of filler used in these paints on the properties of the char formed during the thermal decomposition of the intumescent paint and most important innovative applications of these paints.

## 2. The Effect of Silicone Resins Structure on Their Thermal Stability

The properties of silicone resins differ significantly from those of linear polysiloxanes. The main factors causing these differences are their branched structure and the presence of various types of organic substituents attached to silicon atoms by a Si−C bond. In the synthesis of polysiloxanes, monomers with different degrees of branching are used (see Table 1).

The term ‘‘branched structure’’ means that the polysiloxane contains T or Q units in its chain. The measure of the degree of branching is the ratio of organic groups to silicon atoms (R/Si). The lower R/Si ratio is the higher content of the T units and the branched extent. In the thermogravimetric study of branched and linear polysiloxanes, it was found that thermal stability of branched polysiloxanes was higher as compared to linear ones [20]. The examined branched polysiloxanes were characterized by the degree of branching R/Si ranging from 1.2 to 1.5 and the molar ratio of the phenyl and methyl groups content expressed as moL/moL% ranging from 0/100 to 100/0. The branched polysiloxanes showed a solid residue upon decomposition at 800 °C, depending on the phenyl group content, amounting to 77.3–65.1 wt.% of the initial weight in nitrogen atmosphere and 66.5–40.5 wt.% in air atmosphere. The linear polysiloxane had a degree of branching of 2 and a 75/25 phenyl and methyl ratio, and the solid residue was 37.2 and 26.4 wt.%, respectively, under nitrogen or air. It was found in these studies that the value of the solid residue for a given group of polysiloxanes increased with the decreased content of phenyl groups. However, the solid residue value is not the only criterion for heat resistance. Temperature of degradation is also an important parameter. It is also known that methyl/phenyl branched polysiloxane showed the superior flame-retardant effect as compared to linear polydimethylsiloxane {PDMS) [21,22,23]. Higher thermal stability of methylphenyl polysiloxane as compared to PDMS was confirmed by the measurement of the onset temperature of degradation as nearly 400 °C for methylphenyl polysiloxane compared to 300 °C for PDMS. It is known that the presence of phenyl groups in the resin structure increases their thermostability to 200–250 °C for methylphenyl silicone resin (containing at least 20% of phenyl groups) compared to 180–200 °C for methyl silicone resins [24]. Methylphenyl silicone resins can be used as clear coats or with the addition of inorganic pigments as coating materials for long-term operation at 350 °C.

As shown above, the thermostability of methylphenyl silicone resins depends on their structure expressed by the degree of branching R/Si and on the content of phenyl groups. This gives a very good opportunity to select the resin with the most appropriate parameters necessary to create a good-quality sintered layer created by intumescent paint.

## 3. Formation of Intumescent Layer

The selection of the appropriate type of resin is of great importance for obtaining the desired final effect of fireproof intumescent paints because the process of a protective layer formation is complex and multi-stage, as detailed in Table 2.

For the proper formation of an intumescent layer, it is necessary to use a polymer binder with appropriate thermal stability in order to avoid too early thermal degradation of this binder causing softening and runoff of the paint layer. The thermal degradation of the polymeric binder should take place at a temperature of about 250 °C, enabling incorporation into the sintered layer. The intumescent layer formation process is initiated by the decomposition of ammonium polyphosphate (APP), most commonly used as a source of phosphoric acid, at around 250 °C. The release of H_3_PO_4_ enables further esterification reaction of the hydroxyl groups present in char formers and the polymer binder. A further increase in temperature causes decomposition of esters with the formation of carbon, free acid, water, and carbon dioxide. The decomposition of esters is accompanied by the release of a significant amount of inert gases from the decomposition of blowing agents. The composition and properties of the individual components of the intumescent paint should be selected in such a way that inert gases are released. This causes the protective layer to swell and then harden into char [25]. It can be concluded that obtaining an intumescent layer and ensuring proper protection against fire depends on the correct selection of all ingredients in such a way that the appropriate sequence of their decomposition processes takes place.

## 4. The Effect of Physical Characteristics of Intumescent Layer on the Insulating Properties

Based on the literature data, it should also be noted that the quality of fire protection depends not only on the thickness of the sinter layer. The thermal conductivity of the intumescent layer also has a significant impact on obtaining good protection. Ciprici et al. [26] modelled the properties of intumescent coating using Amon and Denson model [27] to predict bubble growth under pressure in the idealized state conditions. These tests also demonstrated the accuracy of the method of modelling the swelling of coatings by comparing the modelling results under various fire conditions with the results of fire tests performed by Zhang et al. [28,29]. Then, this method was used to model the dependence of the degree of swelling of the coating and its thermal conductivity from coating thickness, the steel thickness and the fire condition including smouldering combustion. It has been found that the expansion coefficient of the intumescent coating decreases, and thus the effective thermal conductivity increases as the heating rate increases. In the modelling studies carried out, it was also assumed that the intumescent layer is multi-layered, which is a very important factor influencing the final result of the determination of the conductivity because individual layers have different thermal conductivities. Such assumptions used in modelling allow to obtain a more accurate result compared to the results obtained for the single-layer model widely used in engineering calculations, including CEN EN 13381-8: 2013 [30].

The effective thermal conductivity of a coating calculated based on standard fire tests should not be used to predict steel temperatures for other heating conditions if the fire is under more severe than standard conditions as the results will not be safe. The results obtained by Ciprici et al. [26] are promising, but require further confirmation and verification for various types of intumescent coatings to be safely used in engineering calculations. Various papers describe the results of numerical modelling assuming equivalent thermal properties of the reactive material [31,32,33]. According to this methodology, the parameters directly measured are the temperature of the steel substrate and the gas temperature or the heat flux released by the cone calorimeter. The values of thermal conductivity of the intumescent layer determined with this method are generally given as a function of temperature [26]. As a solution to this problem Li et al. [34] proposed the use of temperature averaged effective thermal conductivity to determine the temperature history of steel with an applied protective layer. This approach is also often used in engineering calculations, adapted from EN 1993–1-2 [35] where a simple equation links effective thermal conductivity to the temperature increase of steel with respect to a normalized fire curve. Unfortunately, the methodology described above is very imprecise because the influence of a number of phenomena occurring during the creation of the intumescent layer is averaged in effective thermal conductivity as one parameter [36].

Nyazika et al. [37] applied a phenomenological approach to model heat transfer through a silicone-based coating containing expandable graphite exposed to an external heat flux of the order of 50 kW/m^2^. Thanks to the application of the developed model, temperature profiles comparable to the experimental results were obtained. However, the obtained simulations did not capture the temperature increase at times higher than 450 s due to cracking of the intumescent coating [36].

Calabrese et al. [33,36] suggested an innovative experimental methodology based on the use of temperature sensors placed directly inside the expanding intumescent layer and an approximate measurement of the net heat flux by the analyzed structure. This method of determining the thermal parameters of the intumescent layer does not require the definition of free boundary variables of the intumescent layer and the only parameter to be estimated is the apparent thermal conductivity of the reactive material [38]. In order to accurately validate the above methodology, a numerical model was developed in the environment for modern multiphysics simulations (Comsol Multiphysics^®^ 5). The performed tests and numerical simulations confirmed that the heat flow through the system can be evaluated with good approximation. Moreover, it was found that the thermal conductivity of the carbon layer is moderately temperature dependent, while the initial paint thickness does not appear to have a significant effect. Based on the detailed results obtained, the authors estimate that the proposed methodology may be the basis for predict the fire protective capability of intumescent paints. Wang et al. [39] investigated the influence of the intumescent layer structure on thermal conductivity and its changes after accelerated aging. The obtained results of experimental studies were used for validation of an analytical model to calculate the thermal conductivity of the expanded intumescent char. This model was developed using the Russell’s [40] equation for thermal conductivity in porous material. It was found that thermal conductivity was mainly influenced by expansion rate and pore size. Reduced expansion rate and increased pore size after aging result in reduced thermal insulation performance of the intumescent coating. Despite the positive results of the validation of the analytical model, its wider use requires the development of methods to quantify intumescent coating expansion and pore size. It can be concluded that based on the examination of many factors influencing the insulating properties of the intumescent layer, the net heat flux through this layer and the structure of the layer seem to have a significant impact, mainly in terms of its porosity as well as the multilayer structure.

## 5. The Effect of Silicone Resin Structure Used in Intumescent Paints on the Properties of the Char Formed during the Thermal Decomposition of the Intumescent Paint

The unique chemical and physical properties of silicone resins enable their use in various types of intumescent paints formulas, either as the sole binder in the paint or as an additive in the form of a silicone emulsion modifying the properties of typical organic polymers. It is possible to use silicone resin as the only component of the polymer binder because the possibility of selecting a silicone resin with appropriate thermal degradation temperature or thermal stability facilitating the formation of a coating that does not soften too early at elevated temperature. Gardelle et al. [41] studied the properties of intumescent paint prepared by using 100 percent phenyl branched silicone resin having silanol-functionality (6 wt.% of hydroxyl groups). This resin was used in the form of an ethanol solution and the paint formulation additionally contained a modifier consisting of a mixture of PDMS and silica treated with silane. For comparative purposes, waterborne coating for the fire protection of internal structural steelwork which provide up to 120 min fire protection and gives a durable and attractive surface, similar to a paint finish was also tested. The paints were applied on steel plates and dried at 90 °C during 1 h. The obtained coatings were exposed to a flame at a temperature of 1100 °C to obtain charred and swollen layers. It should be emphasized that the carbonized layer from the comparative sample was obtained at a temperature of about 230 °C, while the sample with a pure phenylsilicone resin binder formed such a layer at 380 °C, and with the addition of a modifier at 350 °C. Clear differences were observed in the thickness of the sinter formed during the combustion of the samples. The expansion of the sintered sample based on the phenylsilicone resin with the modifier was about 1000%, while for the sample based on the pure phenylsilicone resin only a negligible degree of expansion was observed. While the sintered layer obtained from the control sample was characterized by 1500% higher expansion compared to the sample based on the phenylsilicone resin with the modifier. It was found that the thermal conductivity of the obtained expanded layers at the temperature of 600 °C was 0.32 + 0.01 W/mK. However, differences were observed in lower temperatures, e.g., at 300 °C for the sample based on pure phenylsilicone resin the value was 0.18 + 0.01 W/mK and for the sample with the modifier 0.13 + 0.01 W/mK. The measurement results at 20 °C showed similar differences for these samples 0.35 + 0.01 and 0.29 + 0.01 W/mK, respectively. The observed differences are probably related to the greater expansion of the sample based on phenylsilicone resin with the modifier and the formation of smaller pores distributed more uniformly throughout the expanded layer structure. On the basis of the thermogravimetric method, it was also found that the modifier based on PDMS degrades twice as fast compared to the phenylsilicone resin, which means that the modifier acts as a blowing agent at the same time. The degradation process of both components takes place at the same temperatures in three stages at 200, 500, and 600 °C. The total weight loss at 800 °C, which is only 31%, indicates a very good heat resistance of both pure phenylsilicone resin and with the addition of a modifier. Based on the analysis of the composition of gases and the solid phase generated in the degradation process, the formation of a cross-linked three-dimensional structure composed of a structure in which four –O–Si bonds are attached to the central silicon, which forms the structure of Q units, was demonstrated. The faster degradation rate of the modifier based on PDMS enables the expansion of a 3D network capable of capturing gases released during the degradation of the modifier [41].

Silicone-based modifiers can also be added to organic binders of intumescent paint to improve their properties [42]. Self-crosslinked silicone acrylate introduced into the paint based on an emulsion epoxy binder allows to obtain intumescent paint not only with fire-retardant but also with anti-corrosive properties [43]. The addition of silicone acrylate has been found to increase the degree of cross-linking of the binder, improving the properties of the coating by reducing water ingress and migration of fire retardants. Based on thermogravimetric tests, it was found that the addition of 14% of self-crosslinked silicone acrylate allows to obtain a homogeneous structure and porosity of intumescent paint without cracks. In these studies, it was also found that increasing the amount of silicone acrylate causes the formation of a large number of large pores, which adversely affects the fire performance.

Very good properties of the intumescent layer have been obtained when using solvent borne silicone-epoxy resin binder [44]. It was found that the dry residue of such paint after annealing at a temperature above 700 °C was clearly higher compared to the intumescent layer of paint with vinyl acetate copolymer dispersion as a binder. The structure of the obtained sintered silicone-epoxy binder was homogeneous, which was important for ensuring adequate protection during a fire. The applicability as a binder of hydroxylated polydimethylsiloxane (PDMS) with a viscosity of 15,000 cSt and methyltrimethoxysilane (MTM) as cross-linking agent was also investigated [45]. When examining the thermal degradation of this binder, it was found that the solid residue after degradation at 800 °C is 2%, which is many times lower value compared to the dry residue after degradation of phenylsilicone resin. [41]. The most important degradation products of PDMS are oligomeric cyclosiloxanes [46]. Therefore, in order to obtain the desired properties of PDMS-based intumescent paint, various fillers should be used, including organoclay, expanded graphite and calcium carbonate.

In conclusion, it can be said that silicone-based binder can be applied for formulation of intumescent paint. Depending on the chemical composition and structure, the properties vary from very good, with the use of branched phenylsilicon resins, to significantly worse for linear polymethylsiloxanes, see Figure 1.

## 6. The Effect of Filler on the Properties of the Char Formed during the Thermal Decomposition of the Silicone-Based Intumescent Paint

Fillers also play an important role in obtaining an intumescent layer with good parameters. Some fillers, due to their chemical composition and structure, have the ability to build into the intumescent layer, positively influencing the properties of the protective layer. Titanium dioxide, commonly used as a white pigment in intumescent paints formulas, plays a special role due to its interaction with other paint ingredients, especially APP. Horacek et al. [47] found that P_2_O_5_ resulting from thermal degradation of APP reacts with TiO_2_ to form titanium pyrophosphate, which is often manifested by the formation of a white foamed layer on the surface of the sintered layer. Moreover, Li et al. [48] found that titanium dioxide in the form of rutile allows to obtain significantly better fire resistance compared to paints containing this pigment in the form of anatase. This effect is probably related to the differences in the size of the TiO_2_ crystallographic domains, which allows them to be packed differently. A similar interaction with the intumescent layer formed of ammonium polyphosphate (APP), pentaerythritol (PER), melamine (MEL) intumescent flame retardant (IFR) system was also found for MoO_3_ and Fe_2_O_3_ [49]. The addition of these metal oxides improves the internal and external structure of the intumescent layer, positively affecting the thermal stability of the layer. Non-reactive fillers such as titanium or zirconium salts [50,51] such as nitrides, borides and carbides [52] also act as thermal stabilizers since they undergo thermal degradation at high temperatures. For example ZrB_2_ due to strong covalent bonding and low self-diffusivity is characterized by high sintering temperatures of over 2000 °C [53].

A significant improvement in the properties of the intumescent layer can also be obtained as a result of the use of fillers capable of reacting with functional groups of binders used in intumescent paints. An example is the use of the popular filler which is chalk for silicone-based intumescent paints. The possibility of reaction between the products of thermal degradation of silicone and chalk is one of the key factors influencing the achievement of a low porosity microstructure of the intumescent layer. These reactions are widely used to create a protective ceramic layer on electric cables with a silicone rubber sheath [54]. The course of chemical reactions taking place during the thermal decomposition of the chalk-containing silicone resin with the formation of calcium silicate, taking into account the diagram proposed by Hermansson et al. [55] is shown in Figure 2.

Obtaining an intumescent layer with good properties is favoured by the formation of calcium silicate with the fibrous structure of wollastonit. Depending on the calcium carbonate content and the thermal decomposition conditions of the silicone resin, calcium silicate with the structure of larnite may also form generally occurring as anhedral to subhedral crystals. Formation of larnite is an undesirable competitive process. Gardelle et al. [45] evidenced based on FTIR and XPS that the top layer of char is composed from Q siloxanes structures and calcium silicate. In the bulk part of the char the presence of other structures such as SiC, SiO_2/2_, SiO_3/2_, and SiO_4/2_ derived from the decomposition of silicone resins was demonstrated. The presence of these structures is of additional importance for good incorporation of the graphite platelets when using expanded graphite as an additional filler. This ensures that a char layer with good cohesion is obtained [56]. Expanded graphite platelets also build into the top layer of the char through interactions with calcium silicate resulting from thermal degradation of chalk increasing the fire performance of the intumescent coating. The obtained results of the critical temperature measurement are a good illustration of the positive effect of the addition of expanded graphite and chalk to the silicone resin binder. For pure steel, the critical temperature of 500 °C is reached after 900 s, while for steel coated with a silicone resin paint with the addition of 25% expanded graphite after 1300 s (±130 s). The addition of graphite allows for some improvement in properties, but the results in terms of stability are worse compared to the commercial paint based on organic resin. On the other hand, the addition of chalk to the paint based on silicone resin and expanded graphite allows for the better fire performance than the commercial intumescent paint, which is expressed by the critical temperature of 465 °C [57]. After introducing 4% by weight of organoclay to the previous recipe char residue exhibits a high expansion rate (3000%), which is similar to the commercial paint. Moreover, the obtained char has good mechanical properties. Further research by TEM, WAXS and 29Si NMR methods on the char structure obtained from silicone resin with the addition of expanded graphite, chalk and organoclay showed that the clear improvement in mechanical properties is related to the strong interactions of organoclay with silicone resin related to the intercalation of some silicate platelets and a chemical reaction between the hydroxyl groups of the organoclay with the silanol groups of the silicone resin with the formation of Si-O-C bonds stabilizing the char structure.

Very positive results of the improvement of the intumescent layer properties were also obtained with the addition of mineral fibres to silicone-epoxy resin based intumescent paint [44]. It was found that the addition of already 3 wt. mineral fibre improves not only the mechanical properties of char but also the anti-corrosion properties, contributing to better protection of steel structures. The effect of different types of fillers on the properties of the char formed during decomposition of the silicone-based intumescent paint is summarized in Table 3.

It can be concluded that fillers have a very significant effect on the properties of the intumescent layer, both in terms of its fireproof properties and other important parameters, including mechanical properties. In recent years, the use of nanofillers for intumescent paints has been intensively studied. This is due to the fact that they often have the ability to build themselves permanently into the intumescent layer. The published results do not directly apply to silicone-based intumescent paints. Nevertheless, it is worth mentioning that positive results of the use of such nanofillers were obtained for organic resins based intumescent paints using clay nano-fillers (layer double hydroxide (LDH), montmorillonite [MMT], and sepiolite) [58], zirconia nanoparticles and chitosan [59] nano -SiO_2_ [60], carbon nanotubes or POSS [61], nano -SiO_2_ and chitosan [62]. Yasir et al. [63] conducted a detailed review of fillers that can be used in organic resins-based intumescent coatings, indicating that obtaining the desired results requires a lot of research and the inclusion of promoters in intumescent paints formulas increasing the possibility of permanent incorporation of fillers into the protective layer. On the basis of the above information on silicone-based intumescent paints, it can also be concluded that the correct selection of fillers in these paints will allow to obtain char layers that meet the highest requirements of fire protection of steel constructions.

## 7. Innovative Applications of Silicone-Based Intumescent Paints

The significant progress in the technology of intumescent paints related to the introduction of new binders with increased thermal resistance, such as, for example, silicone resins capable of reacting with properly selected fillers, contributes to the constant trend of increasing the range of applications of these paints. This trend concerns not only the size of the market, but is also related to the development of new types of paints that give good effects not only on steel but also on other substrates requiring very good fire protection, such as plastics, fabrics, cellulose or wooden products. According to the report Research and Markets [64] the market for intumescent paints was valued at USD 927.6 million in 2018 and is anticipated to progress at a compound annual growth rate [CAGR] of 5.1% from 2019 to 2025. The greatest intensity of growth in the consumption of these paints is observed in the oil & gas sector and the automotive segment, especially in public transport and cargo vehicles. The continued development of the construction sector and large-scale infrastructure developments, especially in developing countries, also has a positive impact on the upward trend. Taking into account the types of paints and the applied binders, it was estimated that the water-based paints sector is developing the fastest, which is due to the increasing VOC limitations. The use of solvent-based paints in low-temperature and high-humidity areas to achieve a decorative finish on complex shapes and to improve adhesion and high water resistance will also increase due to the good properties of these paints. Taking into account the type of binder, the highest growth dynamics is observed in epoxy-based paints. This also presents an opportunity for the increasing the use of silicone resin emulsion binders, which have been shown in several of the publications discussed above to significantly improve the properties of such paints containing hybrid epoxy-silicone-based binders.

In a Goldstein Research [65] report published in May 2020 the trends of the intumescent paints market were similarly assessed. According to this report, the oil and gas industry is the dominant segment of intumescent paints end users, accounting for over 50% of the global market share in 2017. In addition, it is expected to grow by 5% from 2017 to 2030. The fastest growing industry of intumescent paints users is construction with a CAGR of 5.1% for the period from 2017 to 2030, which is caused by large infrastructure projects in China and India. Another area with significant consumption of these paints is the automotive segment, due to the need for a thermal barrier and high temperature protection for the engine in vehicles to ensure increased safety.

It should be noted that the intumescent fireproof coating is one of the easiest and most effective methods of protecting materials, used not only for metal surfaces, but also for plastics, steel, wood, electric cables and polymer composites. This method of protection does not cause chemical modification of the substrate, but rather the formation of a protective layer that changes the heat flux acting on the substrate and may inhibit the temperature of its degradation, ignition or combustion [66]. Even inherently non-combustible steel, when exposed to high temperatures, exhibits a significant reduction in strength and stiffness, which, according to observations made during the collapse of several structures during fires, has a detrimental effect on the stability of the structure [67]. There are a number of types of intumescent coatings on the market. The limitations and recommendations associated with their use should be always considered [68,69]. This should be taken into account in special cases concerning especially paints based on organic polymers. Modification of organic binders with silicone resins improves flame resistance. A number of such solutions are described in patents. So far, some silicone-based intumescent paints [70,71,72] have been patented to significantly extend the fire resistance of the protected substrate. The Korean patent [70] describes the use of a silicone emulsion as an acrylic binder modifier, which improved the properties of intumescent paint. A steel surface protected with the described composition can be kept at the temperature below 649 °C for 2 h. A three-hour resistance time is demonstrated by the silicone resin binder composition described in patent [71], which can be used on rebars, crossbeams, pillars, and ferroconcrete structures in a building. The created intumescent layer shows the maintenance of insulation performances thanks to strong adhesion and wear resistance. Very good results are also reported in patent [72] for intumescent paint with silicone resin binder. Tunnel fire-proof intumescent paint based on an acrylic-silicone binder with very good fire-retardant properties was obtained thanks to the addition of silica nanoparticles [73]. This paint is characterized by the appropriate thickness of the coating layer, adhesion and fire resistance. The intumescent paint based on a silicone acrylic emulsion and the spirocyclic phosphate can be applied to all cellulose textiles [74]. The obtained protective layer is characterized by high mechanical strength, good waterproof and weather resistance and smooth coating surface.

The stability of the carbonized protective layer is crucial to ensuring fire safety in high-rise buildings. It was found that the addition of non-purified fullerene-containing soot with different structure (C60, C70 etc.) and graphite’s microparticles allows to obtain a protective layer with increased strength [75]. The used fillers can have a great influence on the properties of intumescent coatings [76], because the carbonized layer is basically a carbon matrix that can readily accept other carbon materials such as graphite and fullerenes that have proven themselves as reinforcement additives [77], which can protect the charred layer from damage [78].

Intumescent paints are also used to protect plastic surfaces. Beaugendre et al. [79] investigated the properties of protective coatings obtained from a paint containing a mixture of epoxy/silicone resins, a curing agent and either iron oxide or calcium carbonate as fire retardant filler. The obtained results confirmed that the use of these coatings for the protection of polycarbonate allows the creation of a protective barrier limiting the spread of flame, which reduces the flammability of elements made of polycarbonate and reduces dripping of burning material during a fire. It was found that the obtained good properties of the protective layer are related to the incorporation of chalk and iron oxide into the structure of the coating formed during the combustion of epoxy silicone binder. A special intumescent paint creating a flexible and transparent coating on the coated surface is described in the US patent [80]. This composition contains organic resin and/or silicate binders and is designed for the protection of flexible laminates requiring transparent top layers such as electro-optical displays or photovoltaic tiles. Kandola et al. [14] investigated the thermal barrier and fire reaction properties of three commercial intumescent coatings derived from paints containing epoxy binders on glass fiber-reinforced epoxy composites. On the basis of measurements with a cone calorimeter, their thermophysical properties in terms of heating rate and/or coefficient of expansion depending on temperature were determined and correlated with thermal conductivity. The innovative applications of the silicone-based intumescent paints on different substrates showing drawbacks and advantages are summarized in Table 4.

Good durability of the coatings and adhesion between all types of coatings and the substrate were found, indicating the possibility of using intumescent paints to protect glass fiber-reinforced epoxy composites. It can be stated that the protective layers formed from coatings obtained from intumescent paints are increasingly used not only to protect steel structures but also other materials such as building materials and plastics. There are a number of original recipes of these compositions enabling the protection of buildings, e.g., tunnels, vehicle protection and even flexible laminates used in displays in modern electronics.

## 8. Conclusions

The use of silicone resins as binders or their components in intumescent paints is important to obtain the desired parameters of the fireproof layer. The variety of structures of silicone resins and the degree of their branching enables the selection of a binder with appropriate thermal stability so that its thermal degradation takes place at the appropriate temperature at which the sinter layer is formed and that the paint layer does not soften too early, causing it to run off. Based on the literature review, it has also been shown that the parameters of the protective layer are also influenced by the type of organic substituents in the structure of the silicone resin, especially the appropriate content of phenyl groups increasing the thermal stability of the binder. The selection of appropriate fillers has a significant impact on the parameters of the protective layer. Many studies have found that fillers such as chalk, organoclay or expanded graphite have the ability to integrate into the structure of the protective layer formed with the use of silicone resin as a paint binder. This is due to the presence of reactive silanol groups capable of reacting with the functional groups present in the fillers. The properties of the protective layer are also influenced by the structure of the fillers, especially fibrous or layered, which strengthens the mechanical properties of the protective layer. These properties significantly affect the insulation parameters. Based on the examination of many factors influencing the insulating properties of the intumescent layer, it was found that the net heat flux through this layer and the layer structure, mainly in terms of its porosity, as well as the multilayer structure seem to have a significant effect. The material on which the protective layer is placed and their compatibility are also important. The test results correlate well with the results of modelling the insulation of protective layers. Taking into account so many different factors, it seems reasonable to use mathematical modelling, which is a good engineering tool. However, due to the constantly developing area of covering construction and building materials, it is necessary to further develop this tool, taking into account also the scope of fire safety.

The unique properties of intumescent paints, which enable adequate protection of not only steel structures, but also other building materials and plastics, contribute to the constant increase in the use of these paints, thus increasing the safety of people and property during a fire by extending the time necessary to evacuate endangered objects.

Summarizing the information presented and taking into account the architectural tendencies towards increasingly daring, tall forms of steel structures, it should be assumed that the market of intumescent firestop paints will continue to grow intensively. High aesthetics of protective paint coatings and their low weight are advantages inherent in the construction technology of such objects, which are difficult to obtain in other ways. New publications gradually reveal new technical solutions in this field, introducing systems of hybrid paints, intumescent paints with the addition of retardants, new, much more flexible paint binders, as well as new paints using the experience of advanced nanotechnology, allowing for better effectiveness of the fireproof barrier of the coating at lower coating thicknesses.

## Figures and Tables

**Figure 1 materials-13-04785-f001:**
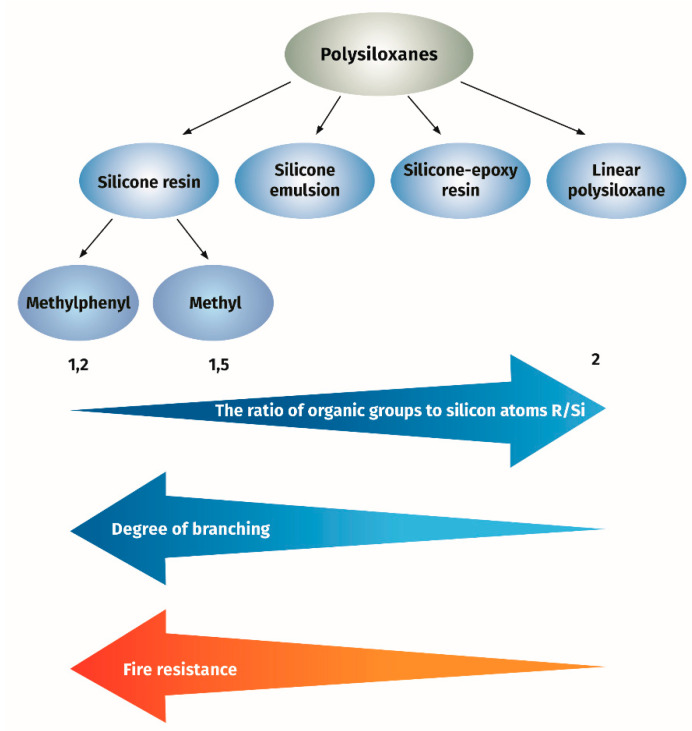
The effect of silicone resin structure on fire resistance of intumescent paint.

**Figure 2 materials-13-04785-f002:**
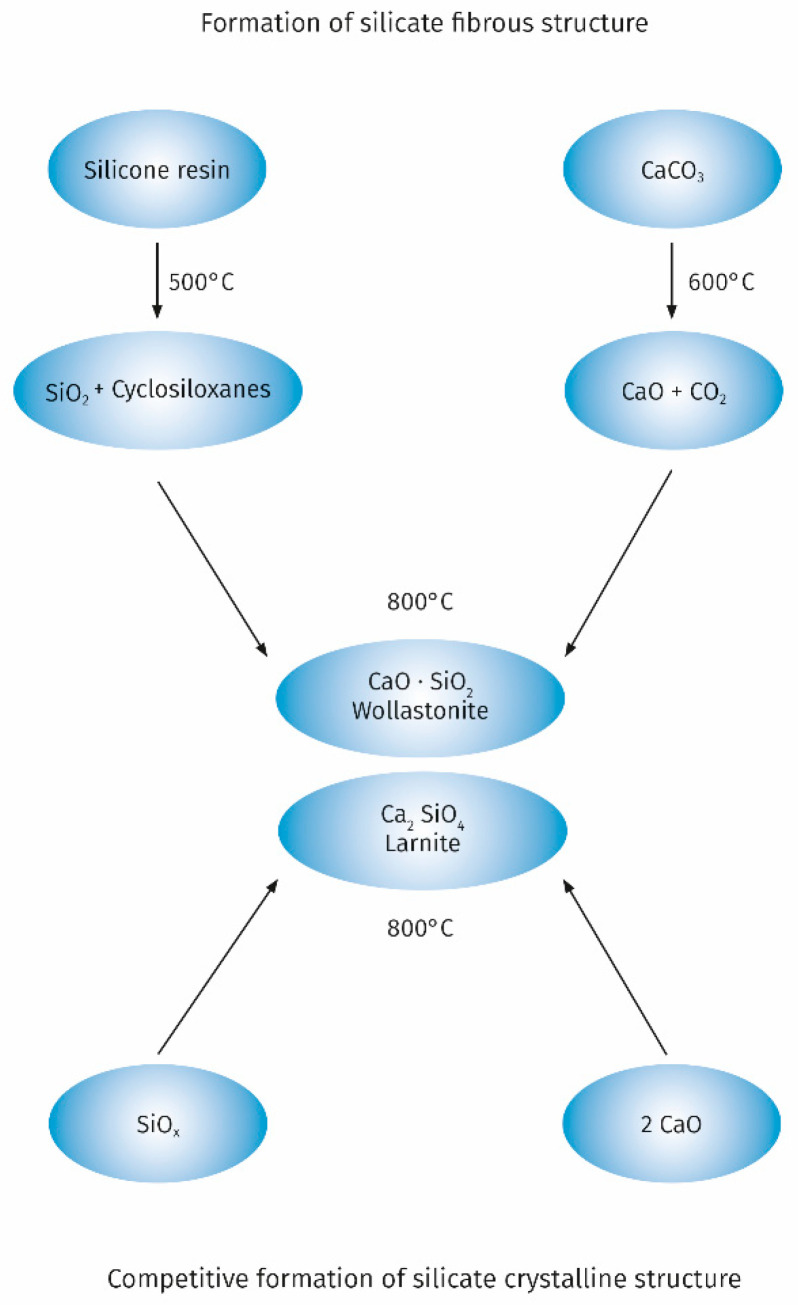
The reactions taking place during the thermal decomposition of the chalk-containing silicone resin.

**Table 1 materials-13-04785-t001:** Monomers with different degree of branching used in the synthesis of polysiloxanes.

Denotation	Stoichiometric Formula	Structural Formula	Degree of Branching
M	R_3_SiO_1/2_	R Me-Si-O- R	1
D	R_2_SiO_2/2_	R -O-Si-O- R	2
T	RSiO_3/2_	R -O-Si-O- O	3
Q	SiO_4/2_	O -O-Si-O- O	4

where R: organic group.

**Table 2 materials-13-04785-t002:** The process of the intumescent layer formation [6,7].

Processes Sequence	Active Component	Role in Intumescent Layer Formation
Preliminary heating	Polymer binder	Softening and melting to ensure the proper properties of the coating
Inorganic acids releasing	Phosphoric acid, its ammonium, aminic salt and esters (ammonium phosphate APP and polyphosphate), boric acid and its derivatives 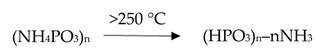	Thermal decomposition of an acid >250 °C which esterifies hydroxyl groups of polymeric binder and char formers
Carbonization of char formers	Polyhydric alcohols (erythritol and its oligomers saccharides and polysaccharides, polyhydric phenols)(HPO_3_)_x_ + C_y_(H_2_O)_z_ 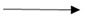 (−C−)_y_ + (HPO_3_)_x_ · zH_2_O	Thermal decomposition results in the formation of carbonaceous material having a large number of hydroxyl groups, able to be esterification with acids
Foam structure formation	Nitrogen or halogen compounds such as melamine and its phosphoric salts, urea, dicyandiamide, guanidine and its derivatives, glycine, chlorinated paraffins 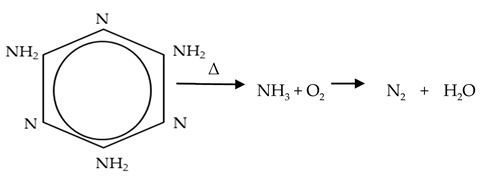	Releases large quantities of non-inflammable gases during thermal decomposition, thus forming foamed structure of carbonaceous layer
Expansion, crosslinking and hardening of the mixture	All components of the paint	Formation of hard carbonaceous layer

**Table 3 materials-13-04785-t003:** The effect of filler on the properties of the char formed during decomposition of the silicone-based intumescent paint.

Type of the Filler	The Role of the Filler Action of the Filler	Drawbacks or Advantages	References
TiO_2_	White pigment interacting with P_2_O_5_ from decomposition of APP giving titanium pyrophosphate	Positive effect on insulating properties due to formation of white foamed layer of titanium pyrophosphate on the surface of sintered layer	[47]
TiO_2_ rutile	More suitable size of crystallographic domains as compared to anatase allowing better packing	Significantly better fire resistance of sintered layer as compared to TiO_2_ anatase	[48]
MoO_3_ and Fe_2_O_3_	Interaction with the components of intumescent layer: APP, PER, MEL	Additives positively affecting thermal stability of intumescent layer	[49]
Titanium or zirconium salts (nitrides, borides, carbides)	Thermal degradation of these salts at high temperatures	Non -reactive fillers acting as thermal stabilizers of the intumescent layer	[50,51,52,53]
CaCO_3_	Thermal decomposition of CaCO_3_ with formation of calcium silicate–wollastonite is preferred due to fibrous structure	Low porosity of the intumescent layer–positive effect on mechanical and insulating properties. Formation of larnite is undesirable competitive process.	[54,55]
Expanded graphite, silicone resins decomposition products, calcium silicate	Graphite platelets build into the char through interactions with silicone resins decomposition products and calcium silicate (decomposition product of CaCO_3_)	Formation of an expanded insulative char (3400% expansion) formed with a high expansion velocity (18%/s) and exhibiting a low thermal conductivity (0.35 W/K m at 500 °C)	[56]
Expanded graphite, organoclay, CaCO_3_	Intercalation of organoclay platelets with chemical reaction between hydroxyl groups of the organoclay and silanol groups of silicone resin with formation of Si-O-C bonds	Stabilization of char structure and clear improvement in mechanical properties of the char	[57]
Mineral fibres, silicone-epoxy resin intumescent paint	Addition of 3 wt.% of mineral fibres which can be incorporated into the structure of the char formed from decomposition of silicone-epoxy resin binder	Improved mechanical properties and anti-corrosion properties. Better protection of steel structures	[44]

**Table 4 materials-13-04785-t004:** The innovative applications of the silicone-based intumescent paints.

Type of Silicone Intumescent Paint	Protected Substrate	Drawbacks or Advantages	Ref
Acrylic binder modified with silicone resin emulsion	Steel	Improved thermal stability. Protected steel surface withstands temperature up to 649 °C for 2 h	[70]
Styrene-modified acrylic copolymer resin containing a silicone-modified acrylic resin that improves fire resistance	Structural elements and reinforced concrete structures	Very good fire resistance (up to 3 h) due to strong adhesin and wear resistance of intumescent layer	[71]
Silicone resin binder obtained by reacting epichlorohydrin with polysilanol compound. Glass flakes added as a filler	Steel and reinforced concrete structures	Can be applied as an alternative material instead of plaster, vermiculite spraying, and concrete fireproof walls	[72]
Acrylic-silicone binder	Tunnel fire-proof protection	The bonding strength is 0.5 MPa, and the fire resistance limit is 90 min.	[73]
Silicone acrylic emulsion containing spirocyclic phosphate intumescent flame retardant	Cellulosic textile	Long burning resistance time, high mechanical strength, good waterproof and weather resistance and smooth coating surface. A preparation technology is simple in process and convenient in operation	[74]
Silicone resin binder, non-purified fullerene soot, graphite microparticles	High-rise buildings	Reinforced structure of intumescent layer	[75,76,77,78]
Epoxy-silicone binder, CaCO_3_ or Fe_2_O_3_	Polycarbonate	Formation of intumescent layer limiting the spread of flame. Reduced dripping of burning material during a fir	[79]
Epoxy-silicone resin binder, CaCO_3_, Fe_2_O_3_	Electro-optical displays or photovoltaic tiles	Formation of flexible and transparent coating	[80]
Aqueous organic polymer dispersion and silicone resin emulsion	Steel, aluminium, wood, concrete, electric cables and pipes, or for coating open steel profiles, closed and/or castellated profiles, or for workshop applications.	Easy to application intumescent paint forming protective layer characterised by very good mechanical and thermal properties	[81]
Multilayer epoxy resin based intumescent paint. Silicone resin as binder for the top layer	Different surfaces metal, wood, plastics used in construction	The disadvantage is the necessity to apply the paint twice. The first layer is based on epoxy resin and the second layer is based on silicone resin	[82]
One or more crosslinkable silicone polymer	Applicable to products formed for fire wall linings, fire partitions, screens, ceilings or linings, structural fire protection, fire door inserts, window or door seals, intumescent seals, in electrical switchboard cabinets or cables. In one cable application, the composition may be used as the extruded intermediate material	Very good mechanical properties and high temperature stability	[83]
Intumescent composition comprising a polydimethylsiloxane (A) of degree of polymerisation at least 300 siloxane units and expandable graphite and a titanate catalyst	Protection of a metal, wood or plastics substrate exposed to fire risk from hydrocarbons	Reinforced structure of intumescent paint. Very good thermal insulation and adhesion to different substrates	[84]

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
