# Peer review of "Silicone Resin-Based Intumescent Paints"

_materials, 2020, doi:10.3390/ma13214785_

Round 1
Reviewer 1 Report
Dear authors,
your paper intitled "Silicone resins-based intumescent paints" and submitted to Materials journal of MDPI must be revised.
Major revisions:
- Introduction section: from #30 to #112 there are some negligible parts. Please summarize.;
- The authors should underline the drawbacks and the advantages of thematerials used in the previous works/studies. Please use a table (or two tables), in order to make clear the differences (positive or negative) between materials. For example: one table for the effect of filler with comments (drawbacks and advantages) and relative references, one table for the innovative applications with comments and relative references (drawbacks and advantages);
- Conclusions section: please try to summarize.
Minor revisions:
#112: error in the unit of measure;
#124: "Materials The properties..". Check!;
#129: space between the table 1 and the line #129. Check for the other tables, too.
Best regards.
Author Response
Response to Reviewer 1 Comments
We would like to thank the Reviewer for careful and thorough reading of this manuscript and for the thoughtful comments and constructive suggestions, which help to improve the quality of this manuscript. Our responses are as follows:
General comment:
Your paper intitled "Silicone resins-based intumescent paints" and submitted to Materials journal of MDPI must be revised.
We appreciate this comment while making every effort to ensure that the corrected version has the best possible quality.
Point 1: Introduction section: from #30 to #112 there are some negligible parts. Please summarize.;
According to Reviewer the negligible parts were canceled and the summary sentences were added.
Point 2: The authors should underline the drawbacks and the advantages of the materials used in the previous works/studies. Please use a table (or two tables), in order to make clear the differences (positive or negative) between materials. For example: one table for the effect of filler with comments (drawbacks and advantages) and relative references, one table for the innovative applications with comments and relative references (drawbacks and advantages);
According to Reviewer the following two tables were added:
- the table concerning the effect of filler on the properties of the char formed during decomposition of the silicone-based intumescent paint containing drawbacks and advantages of the applied filler as well as relative references;
- the table concerning the innovative applications of the silicone-based intumescent paints on different types of substrates. The positive and negative comments with the appropriate references (also new ones) were included.
Point 3: Conclusions section: please try to summarize.
According to Reviewer 1 comment 3 Conclusions were restructured and summarized.
Point 4: Minor revisions:
#112: error in the unit of measure;
#124: "Materials The properties..". Check!;
#129: space between the table 1 and the line #129. Check for the other tables, too.
All minor revisions were done according to Reviewer’s recommendations.

Reviewer 2 Report
The review covers the area of thermo protective intumescent paints based on silicone intended for steel structures. It is very well written and informative. I recommend publication after the following minor additions/corrections.
1) Since the authors are referring to “various types of organic substituents” I would suggest replacement of “Me” in Table 1 with R.
2) Page 4 Lines 143-144: The abbreviation PDMS (polydimethylsiloxane I suppose) must be defined before use. Furthermore it does not fit to a methyl/phenyl branched polysiloxane.
3) Since the process of the intumescent layer formation quoted in Table 2 seems very interesting the inclusion of indicative chemical reactions taking place in the various stages would greatly enhance the quality of the manuscript.
4) There is an error in Figure 1. Polysiloxanes can be either linear or branched. The diagram as is, implies that linear polysiloxanes are a subcategory of branched and is misleading. Furthermore the term “silicon-epoxy” puzzles me. Are the authors referring to silicon-epoxy resins?
Author Response
Response to Reviewer 2 Comments
We wish to thank Reviewer for the constructive comments providing valuable insights to the contents of this manuscript. We tried to address the issues raised as best as possible. Our response follows:
General comment:
The review covers the area of thermo protective intumescent paints based on silicone intended for steel structures. It is very well written and informative. I recommend publication after the following minor additions/corrections.
The authors appreciate very much this comment.
Point 1: Since the authors are referring to “various types of organic substituents” I would suggest replacement of “Me” in Table 1 with R.
According to Reviewer comment “Me” were replaced with “R”.
Point 2: Page 4 Lines 143-144: The abbreviation PDMS (polydimethylsiloxane I suppose) must be defined before use. Furthermore it does not fit to a methyl/phenyl branched polysiloxane.
According to Reviewer comment the nomenclature of polysiloxanes was rearranged.
Point 3: Since the process of the intumescent layer formation quoted in Table 2 seems very interesting the inclusion of indicative chemical reactions taking place in the various stages would greatly enhance the quality of the manuscript.
According to Reviewer comment the indicative chemical reactions taking place in the various stages of the intumescent layer formation were included in the Table 2.
Point 4: There is an error in Figure 1. Polysiloxanes can be either linear or branched. The diagram as is, implies that linear polysiloxanes are a subcategory of branched and is misleading. Furthermore the term “silicon-epoxy” puzzles me. Are the authors referring to silicon-epoxy resins?
According to Reviewer comment the new version of Figure 1 was added. The description “Branched polysiloxanes” has been replaced with the word polysiloxanes. The new description “silicone epoxy resins” was added instead of “silicone epoxy”.

Reviewer 3 Report
Dear Authors,
the topic of fire protection is very important in our daily life and in a lot of fields. In consequence, it is always interesting to read an updated review on the subject. The boundary of the topic and the methodology (keywords and databases) used to choose the different articles for this review are clearly presented in the introduction. Nevertheless, some points are confusing for the reader.
The title of this review is "Silicone resins-based intumescent paints". With this title, the reader expects to find information about silicone-based paints to protect a wide variety of materials. Although we can find minor results on cellulose textiles or plastics at the end of the paper, all the abstract, introduction and a large part of the paper are dedicated to "fire insulating" of steel or aluminium structure. The title should be modified or the paper should be modified to match.
The construction of the different parts is also confusing. For example, it is specified that part 3 is dedicated to the formation of the intumescent layer. It is effectively the case of the beginning of this part. However, from the line 173, the link between the text and the title of this part is not very clear. In my opinion, from this line, the text describes more some physical characteristics of the layer (thickness, thermal conductivity) than the formation of this one.
For a review, I consider that there is a lack of tables or graphs synthesizing the fire properties of the different silicon-based intumescent paints. Although true, the remarks are too general and do not focus on the specific steel protection specifications (except for line 431-432. As a reader I want to find this kind of information in such specific review).
Finally, there are minor typing errors and figure 1 should be modified because it is not understandable as well.
For all these reasons, I recommend reorganizing deeply this review before a possible publication.
Author Response
Response to Reviewer 3 Comments
We would like to thank Reviewer for the detailed comments providing valuable insights to the contents of this manuscript. We tried to address the issues raised as best as possible. Our response follows:
General comment
The topic of fire protection is very important in our daily life and in a lot of fields. In consequence, it is always interesting to read an updated review on the subject. The boundary of the topic and the methodology (keywords and databases) used to choose the different articles for this review are clearly presented in the introduction. Nevertheless, some points are confusing for the reader.
We appreciate this comment very much. We while making every effort to ensure that the corrected version has the best possible quality. The Reviewer's comments will help us achieve the correct level of the revised version.
Point 1: The title of this review is "Silicone resins-based intumescent paints". With this title, the reader expects to find information about silicone-based paints to protect a wide variety of materials. Although we can find minor results on cellulose textiles or plastics at the end of the paper, all the abstract, introduction and a large part of the paper are dedicated to "fire insulating" of steel or aluminium structure. The title should be modified or the paper should be modified to match.
According to Reviewer comments a sentence was added specifying more precisely the content of the paper. Moreover. in the Table 4 concerning the innovative applications of the silicone-based intumescent paints some new examples of the intumescent paints application on the different substrates were added based on new references to patents descriptions.
Point 2: The construction of the different parts is also confusing. For example, it is specified that part 3 is dedicated to the formation of the intumescent layer. It is effectively the case of the beginning of this part. However, from the line 173, the link between the text and the title of this part is not very clear. In my opinion, from this line, the text describes more some physical characteristics of the layer (thickness, thermal conductivity) than the formation of this one.
According to Reviewer comment the text of part 3 was divided into two parts and the second part concerning physical characteristics of the layer was entitled: The effect of physical characteristics of intumescent layer on the insulating properties
Point 3: For a review, I consider that there is a lack of tables or graphs synthesizing the fire properties of the different silicon-based intumescent paints. Although true, the remarks are too general and do not focus on the specific steel protection specifications (except for line 431-432. As a reader I want to find this kind of information in such specific review).
According to Reviewer the table concerning the innovative applications of the silicone-based intumescent paints was included. The advantages and disadvantages of each paint with the appropriate references (also new ones) were included.
Point 4: Finally, there are minor typing errors and figure 1 should be modified because it is not understandable as well.
The modified version of Figure 1 was added.
Point 5: For all these reasons, I recommend reorganizing deeply this review before a possible publication.
We hope that all done corrections and additions improved the quality of the manuscript.

Round 2
Reviewer 1 Report
Dear authors,
your paper intitled "Silicone resins-based intumescent paints" is now accepted.
Check for the metrics of the Materials Journal MDPI before the final submission.
Best regards and take care.
Reviewer 3 Report
Dear Authors,
you address all my remarks and make the necessary modifications.
I consider that the paper can now be accepted in its present form for publication.